# Impacts of *Parthenium hysterophorus* L. on Plant Species Diversity in Ginir District, Southeastern Ethiopia

**Mesfin Boja** [1,*], **Zerihun Girma** [2] **and Gemedo Dalle** [3]

1    Ethiopian Biodiversity Institute, Goba Biodiversity Center, Goba, Addis Ababa P.O. Box 30726, Ethiopia
2    Department of Wildlife and Protected Area Management, Hawassa University,
     Awassa P.O. Box 128, Ethiopia
3    Center for Environmental Science, College of Natural and Computational Sciences,
     Addis Ababa University, Addis Ababa P.O. Box 1176, Ethiopia
*    Correspondence: mesfinboja3@gmail.com

**Abstract:** Invasive alien species are considered the second greatest global threat to biodiversity. This study is aimed at determining the impacts of *Parthenium hysterophorus* on herbaceous and woody plant species diversity in the Ginir district, southeast Ethiopia. Data on vegetation were collected from the three study sites' four land use types, with each land use type having invaded and non-invaded land units. A systematic random sampling method was used for establishing sampling plots. To examine the impacts of the invasive on native plant diversity, a total of 160 plots (120 plots of 1 $m^2$ on grazing lands, roadsides, and abandoned agricultural lands and 40 plots of 20 $m^2$ for sampling herbaceous and tree (shrub) species, respectively) were established. The number of plots was equally distributed in both invaded and adjacent non-invaded areas. Plant species from each plot were recorded and identified. In each plot, all the individuals of *P. hysterophorus* were counted, the heights of the five tallest individuals were measured, and the mean height was calculated. The percentage cover of *P. hysterophorus* was visually estimated. The data were analyzed using both descriptive and inferential statistics. A total of 105 plant species (45 trees/shrubs and 60 herbaceous) belonging to 84 genera and 42 families were documented in the study area. The result showed a strong negative relationship between the density of *P. hysterophorus* and other plant species richness (r = −0.82, *p* = 0.013) and species abundance (r = −0.917, *p* = 0.001) per study site of the invaded community. Species richness in the non-invaded site was higher (105 species) than in the invaded area (63 species), demonstrating the negative impact of *P. hysterophorus* on local biodiversity. Furthermore, the number of plant families was 42 in the non-invaded area, in contrast to only 32 in the invaded areas, a 23.8% decline. Of the plant communities, similarity indices between non-invaded and invaded sites among different land use types were >50%. It was concluded that *P. hysterophorus* was one of the most dominant invasive alien species in the study area that reduced the species diversity of various plant species. Putting in place a strategy and effective planning for the control and management of this invasive alien species is strongly recommended.

**Keywords:** alien species; IAS; invasive species; land use types; non-invaded; Santa Maria feverfew; weed; woody plants

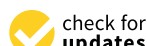



## 1. Introduction

Invasive alien species (IAS) are the key drivers behind the current biodiversity loss. It is also among the evil five collectively known threats to biodiversity, i.e., habitat modification, over-exploitation, climate change, and chains of extinction [1,2]. Thus, biological invasions attract the concern of conservationists, ecologists, foresters, policymakers, and other scientists. This is due to their potential to cause hundreds of biological extinctions throughout the world, which have a great influence on society, economic life, health, and national heritage [3,4].

Since the 17th century, IAS has been responsible for nearly 40% of all known animal extinctions [5]. The estimated economic damage from invasive species worldwide has been USD 1.288 trillion over the past 50 years [6], close to 5% of global growth domestic product (GDP), with impacts across a wide range of sectors, including agriculture, forestry, aquaculture, transportation, trade, power generation, and recreation [7,8].

A 2021 study in North America estimated that invasive species cost USD 2 billion per year in the early 1960s and has increased to over USD 26 billion per year since 2010 [9]. The total economic costs incurred by the IAS on the Indian economy ranged from USD 137.3 billion to 182.6 billion between 1960 and 2020 [10]. In particular, the problem of invasive weeds is particularly serious in the poorest and most vulnerable countries. In Southeast Asia, invasive species cost at least USD 33 billion per year, accounting for 5% of total GDP [11]. Accordingly, the management of invasive species is one of the strategic intervention areas in achieving the Sustainable Development Goals (SDGs) Goal 2, which is to end hunger, achieve food security and improved nutrition, and promote sustainable agriculture [12].

Africa may be particularly vulnerable to exotic and invasive species' colonization. This may be due to its climate-sensitive distribution of native flora and fauna [13]. As biological invasion attracts both scientific and political attention [14], Ethiopia recognized the threats posed by invasive alien species (IAS) to local biodiversity and incorporated this fact into its various policy and strategy documents.

In Ethiopia, IAS are causing a variety of issues for agricultural lands, range lands, biodiversity, national parks, waterways, rivers, power dams, roadsides, and urban green spaces, with serious economic and environmental consequences. Approximately 35 invasive alien species were recorded, with *Parthenium hysterophorus* being among the top five highly targeted weeds [15,16].

*Parthenium hysterophorusis* (Santa Maria feverfew) is an annual herb in the family Asteraceae that is characterized by a deep taproot, pale green leaves, and an erect stem that gradually becomes woody. At maturity, the plant develops several branches in its top half and may finally reach a height of 1.5–2 m (European and Mediterranean Plant Protection Organization [17]. The genus *Parthenium* has 16 species native to subtropical areas in northern South America, Central America, Mexico, Texas, and Florida but is currently widely distributed in tropical and subtropical countries such as Australia, China, Kenya, Ethiopia, Israel, Taiwan, India, and Nepal and has invaded as many as 30 countries around the globe [18,19].

The dispersal of *P. hysterophorus* occurs in multiple ways, including short-distance wind dispersal, or water surface, runoff in natural streams and rivers, in irrigation and drainage channels, and irrigation water from the ponds or through farm machinery, vehicles, movement of livestock, animal dung, and grain seeds [20,21]. In Ethiopia, the invasion history of this weed is not well known. However, anecdotal shreds of evidence indicate that it is widely distributed throughout the country even though there is no actual baseline data. It was believed to have been accidentally introduced to Ethiopia from North America in the 1970s when drought-induced famine triggered a massive multinational relief effort. It was supposed to be introduced as a contaminant of grain food aid and distributed with the grain [22].

*P. hysterophorus* is a major weed, existing in more than 45 countries [23,24]. It has been reported to have socio-economic impacts, including a decline in crop and livestock production, human health, soil fertility, and biodiversity. This invasive species changes the scenario of the agriculture of the world, i.e., reducing crop production by forty to fifty percent and pasture production by up to 90% [25]. For instance, in Australia, the economic damage caused by this species every year has been estimated to be USD 16.5 million in the beef industry and several million dollars in the cropping industries [26]. In eastern Ethiopia, the yield of sorghum grain was reduced by 40% to 97% [27] and by 18.8–86.4% in the common bean when *P. hysterophorus* was left uncontrolled throughout the season [28].

Data on the impacts of *P. hysterophorus* on the species richness, evenness, diversity, and composition of invaded communities is limited [29]. A few studies conducted in different parts of Ethiopia have revealed the aggressiveness of *P. hysterophorus* species on native plant species [16,30,31]. However, there has not been a specific study on the impacts of *P. hysterophorus* on the indigenous plant diversity in southeastern Ethiopia. In particular, the specific study attempted to answer previous, less explored research questions such as how *P. hysterophorus* affects native plant species composition, richness, and abundance and how it adversely affects native floral community structure in a specific agro-ecological condition. In order to fully understand the impact of this IAS on native biodiversity and take likely control measures, there is a need for proper documentation of its impacts in different agro-ecologies. This research is therefore aimed at determining the impacts of *P. hysterophorus* on the herbaceous and woody plant species diversity of invaded land use types in Ginir woreda of the East Bale zone, Oromia regional state, Ethiopia.

## 2. Materials and Methods

### 2.1. Description of the Study Area

The study area, Ginir district, is found in the East Bale zone, Oromia Regional State, Southeast Ethiopia. It is one of the administrative units (woreda) among the 7 districts of the East Bale zone, with an area of approximately 2384 km$^2$. Ginir town is the administrative center of the district, which is located at a distance of 136 km from the Bale zone capital, Robe town, and 566 km from the country's capital, Addis Ababa (Figure 1).

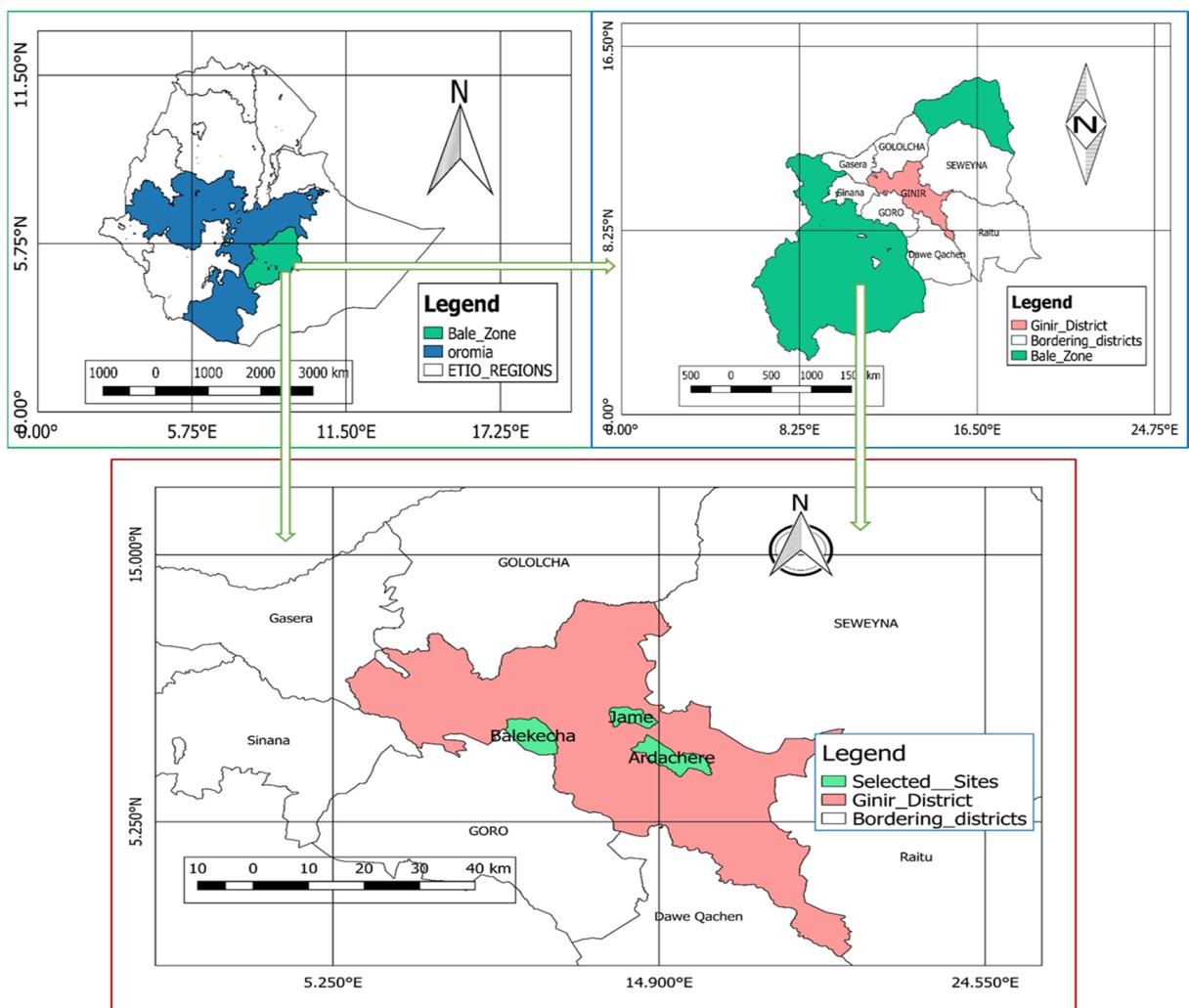

**Figure 1.** Location map of the study area.

According to the Central Statistical Agency's population projection, the total population of the district by the year 2021 was estimated to be 203,751 (103,592 males and 100,159 females). The topography of the district falls within the altitudinal range of 1200–2406 m above sea level. According to data from the district agricultural office, the land configuration of the district is categorized as plain, which accounts for approximately 85%, mountain 3%, and rugged and gorge areas account for approximately 12% (i.e., approximately 15% of the area of this district is covered with a valley, gorges, and hills). Similarly, the land use in the district indicates that 30.5% is arable or cultivable, 31.2% is pasture, 35.6% is forest, and the remaining 2.7% is considered swampy, mountainous, or otherwise unusable.

### 2.2. Methods

#### 2.2.1. Sampling Design

Based on data from the Ginir district agricultural office and visual observation, three kebeles (smallest administrative units), namely Ardatare, Balekecha, and Jamie, were purposefully chosen. A systematic random sampling method was used to establish sampling plots to cover the total sampling sites [32]. This helped to include as many vegetation types as possible to represent the native flora of the study area. For each sampling area, line transects were used in a way that incorporated a sufficient number of sample plots from both invaded and non-invaded areas. This was conducted by applying the plot method [33] to sampling.

Sample plots were established in both invaded and non-invaded areas to compare the plant species diversity. In each sampling plot, one plot of the pair was placed in *P. hysterophorus* invaded plots, where there was a high infestation, and the second plot was placed in neighboring vegetation, where there was no infestation [34]. The plots were chosen to cover a range of site conditions and vegetation types in which the invader achieves dominance in the invaded communities. In a few cases, very low and recently emerging stems of the *P. hysterophorus* occurred in the non-invaded plot, which may not have induced any changes to vegetation structure and species composition [35]. As much as possible, the non-invaded plots were chosen to have similar site conditions (10 m) to the invaded plots [36].

Four dominant land use types were selected using a stratified random sampling method, i.e., woodland, grassland, roadside, and abandoned agricultural land. Four infested *P. hysterophorus* patches from each land use type of the study sites were selected and numbered, and two patches from each land use type with an area of >one hectare [37] were randomly chosen from each list of patches. The level of *P. hysterophorus* invasion and the size of the area covered were selected to increase the efficiency of sampling due to the assumption that areas of high *P. hysterophorus* cover and invasion could have a high impact.

To collect herbaceous vegetation data from grazing lands, roadsides, and abandoned agricultural lands, a hundred meter (100 m) long transect was established in each sampling site (patch) and along it, ten 1 m × 1 m (1 m$^2$) size plots were systematically laid at an interval of 5–10 m [38,39]. To allow equal chance sampling between invaded and non-invaded areas, the number of plots was equally distributed in both invaded and adjacent non-invaded areas. As a result, depending on the field situation, data from non-invaded plots were collected at a distance of 10 m from invaded plots [36]. Thus, forty plots (20 plots for invaded and the rest 20 plots for non-invaded) were sampled from the different sample sites for each land use type. Therefore, a total of 120 plots were sampled for the three land use types in each of the three kebeles (Figure 2).

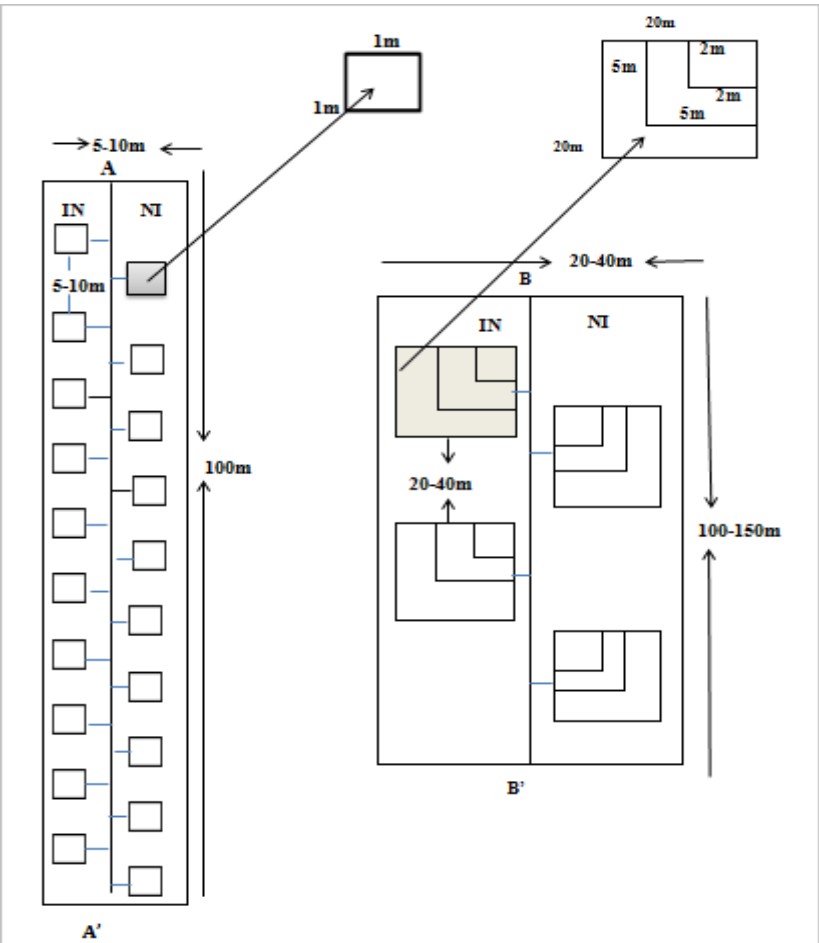

**Figure 2.** Layout of sample plots for vegetation data collection in different land use types. **Notes**:—Transect A–A′ shows layout for herbaceous vegetation data collection in agricultural land, roadside, and grassland land use types (10 similar layouts of transects were laid in each site). Transect B–B′ indicates layout for vegetation data collection in woodland land use types (five similar layout transects were laid in each site).

To examine the impacts of *P. hysterophorus* invasions on the indigenous woody species diversity of invaded communities, sampling plots were located purposively in two of the three kebeles based on the relatively high magnitude of infestation in the woodland areas of these kebeles. In each of the two sampling sites, five equally drawn imaginary transects were defined at an interval of horizontal distance of 100 to 150 m, passing along the length of the respective woodland plant community. In each of the five transects, two plots of 20 m × 20 m were taken for tree sampling, with nested plots of 5 m × 5 m (saplings and shrubs) and 2 m × 2 m (seedlings) taken within each sample plot [40]. To sample trees and shrubs, seedlings, and saplings of woody plants, ten plots were laid on invaded woodland forest areas and the remaining half in adjacent non-invaded areas, for a total of 20 plots in each sampling site (Figure 2).

### 2.2.2. Data Collection

Vegetation sampling was conducted during November and December 2019. The study assessed the impacts of *P. hysterophorus* on herbaceous and woody species diversity in different infested land use types. Each native plant species was identified and recorded. In each plot, all the individuals of *P. hysterophorus* were counted. The heights of the five tallest individuals were measured, and a mean height was calculated. The percentage cover of the invasive was visually estimated by classifying it into different infestation levels in

each plot using the procedure documented in [33] to determine its impact on indigenous plant diversity.

Sample specimens were taken to the Ethiopian Biodiversity Institute Herbarium for identification and proper naming. The collected specimens were identified using authenticated specimens, consulting experts, and referring to the eight published volumes of *Flora of Ethiopia and Eritrea* [41,42].

2.2.3. Data Analysis

Both descriptive (frequency and percentage) and inferential statistics were used for data presentation and analysis. The biophysical data were organized using "Microsoft Office Excels". Data collected from the plots survey was analyzed using different statistical tools with the help of IBM SPSS (Statistical Program for Social Sciences) Statistics for Windows Version 20.0. Armonk, NY: IBM Corp. IBM Corp. Released 2012 software.

All the plant species identified in this study were ranked according to their family. The diversity of the species for the vegetation data from the sample sites was compared using the Shannon Diversity Index [43]. This index accounts both for the abundance and for the evenness of the species in the natural environment. Abundance is the total number of occurrences of species in each plot across the different land use types, and evenness explains how equally abundant each species would be in the plant community [44].

The evenness of species was calculated as described by [45]. The species evenness has values between 0 and 1, where 0 indicates the abundance of a few species and 1 indicates the condition where all species are equally abundant. Besides, high evenness is a sign of ecosystem health [46]. A community with a high even-ness index is characterized by a large number of species that are distributed equally in most sample plots, and a community with a high evenness index is more stable than the lesser ones [47].

This was also used to assess the impact of *P. hysterophorus* on the diversity of herbaceous, shrub, and tree plant species. The higher value of the index of diversity indicates the variability in the type of species and heterogeneity in the community, whereas the lower values point to the homogeneity in the community.

The frequency of each species in non-invaded and invaded sites was calculated. Jaccard's Similarity Index and Sorenson's Similarity Index between non-invaded and invaded areas for the selected land use types of each sampling site were calculated [48,49].

Plot-wise data of vegetation attributes were used in the statistical analysis. A linear regression and correlation analysis were used, with the *P. hysterophorus* density as the independent variable and species richness, species abundance, and height of *P. hysterophorus* as the dependent variables. A multicollinearity test among the independent variables was carried out by the evaluation of linear regression and Pearson correlations. The regression equation was computed as $y = b_0 + b_1x_1 + b_2x_2 + b_3x_3$, where $y$ is *P. hysterophorus* density, $x_1$ is species abundance, $x_2$ is species richness, and $x_3$ is the height of *P. hysterophorus* per study site, $b_0$ = regression constant and $b_1$, $b_2$, $b_3$ are estimated regression coefficients.

## 3. Results

### 3.1. Impacts of Parthenium on Herbaceous and Woody Species Diversity

The result showed that there were 105 species in the non-invaded area and 63 species in the invaded sites (Appendix A). Thus, the number of species was reduced by 40% in the *P. hysterophorus* invaded area as compared to the non-invaded area. According to the result, the vegetation invaded by the invasive has fewer species in herbaceous and woody plant species diversity compared to the non-infested vegetation over all the land use types.

The results were expressed as mean ± standard deviation of the mean (SEM). The results of the Shannon–Wiener diversity index showed that the non-invaded part of the woodland area has relatively the highest mean species diversity (3.38 ± 0.1365) and richness (44.5 ± 2.5). On the contrary, the least mean species diversity (1.724 ± 0.045) and richness (9 ± 1.0) were recorded in the *P. hysterophorus* invaded vegetation along roadsides and abandoned agricultural land, respectively (Table 1).

**Table 1.** Mean species abundance, richness, evenness, and diversity for invaded and non-invaded land units among different land use types in Ginir district.

| S.N | Kebeles | LUT | Abundance | | Richness | | Evenness | | H′ | |
|---|---|---|---|---|---|---|---|---|---|---|
| | | | IN | NI | IN | NI | IN | NI | IN | NI |
| 1 | Balekecha | Wood land | 111 | 265 | 27 | 42 | 0.865 | 0.869 | 2.85 | 3.247 |
| | Ardatare | Wood land | 122 | 276 | 29 | 47 | 0.895 | 0.914 | 3.016 | 3.52 |
| | | Mean | 116.5 ± 0.5 | 270.5 ± 0.5 | 28 ± 1.0 | 44.5 ± 2.5 | 0.88 ± 0.015 | 0.891 ± 0.0225 | 2.93 ± 0.083 | 3.38 ± 0.1365 |
| 2 | Balekecha | Roadside | 138 | 231 | 10 | 17 | 0.729 | 0.802 | 1.68 | 2.27 |
| | Ardatare | Roadside | 72 | 133 | 8 | 11 | 0.85 | 0.77 | 1.77 | 1.849 |
| | | Mean | 105 ± 0.3 | 182 ± 1.9 | 9 ± 1.0 | 14 ± 3.0 | 0.789 ± 0.6 | 0.786 ± 0.016 | 1.724 ± 0.045 | 2.059 ± 0.21 |
| 3 | Jamie | A. agri. land | 148 | 267 | 8 | 12 | 0.955 | 0.921 | 1.986 | 2.28 |
| | Ardatare | A. agri. land | 128 | 250 | 10 | 11 | 0.961 | 0.967 | 2.21 | 2.27 |
| | | Mean | 138 ± 1.0 | 258.5 ± 1.5 | 9 ± 1.0 | 11.5 ± 0.5 | 0.957 ± 0.003 | 0.944 ± 0.023 | 2.098 ± 0.112 | 2.275 ± 0.005 |
| 4 | Ardatare | G. land | 672 | 1296 | 19 | 27 | 0.74 | 0.688 | 2.18 | 2.27 |
| | Jamie | G. land | 907 | 2108 | 17 | 29 | 0.787 | 0.734 | 2.229 | 2.47 |
| | | Mean | 789.5 ± 1.17 | 1702 ± 2.6 | 18 ± 1.0 | 28 ± 1.0 | 0.763 ± 0.0235 | 0.711 ± 0.023 | 2.204 ± 0.0245 | 2.37 ± 0.1 |

**Note:** LUT (Land use types), IN (Invaded area), NI (Non-invaded), A. agri. land (Abandoned agricultural land), G. land (Grassland).

Regarding the highest mean evenness value of the study sites, approximately similar results were recorded for both invaded (0.957 ± 0.003) and non-invaded (0.944 ± 0.023) land units in abandoned agricultural lands. On the other hand, the least mean evenness values were noted in the grazing lands (Table 1). On the other hand, the mean evenness value of the entire invaded sampled study sites was 0.847, indicating 84.7% of the plant communities had a uniform distribution, while the mean evenness value of the non-invaded samples was 0.833, indicating 83.3% of the plant communities had a uniform distribution. Thus, the heterogeneity of the invaded study sites was reduced by a very small value, that is, 1.4%.

### 3.1.1. Impacts of *P. hysterophorus* Invasion on Species Composition

The results of this study showed that the invasion of *P. hysterophorus* severely affected the composition of vegetation in the study area. Twenty-seven and forty-two plant species were recorded from each invaded and non-invaded land unit of the Baliekecha woodland sampling plots, respectively, whereas twenty-nine and forty-seven plant species were recorded in both invaded and non-invaded plots of the Ardatare woodland site. Ten and seventeen plant species were recorded from invaded and non-invaded plots of Baliekecha roadside plots, respectively (Table 1).

A total of 45 tree (shrub) and 60 herbaceous plant species belonging to 84 genera were documented from the study sites of different land use types. The number of plant families was 42 in the non-invaded area, in contrast to only 32 in the invaded areas. In non-invaded areas, the family Fabaceae was represented by the highest number of species (17 species), accounting for 16.35%, followed by Poaceae (13.46%), Asteraceae (7.69%), Euphorbiaceae (5.77%), and Bursseraceae (3.85%). It is worth noting that the above-mentioned five families alone represent the bulk of (47.12%) plant species in the total flora in non-invaded (controlled) study sites (Appendix A).

The plant species in the non-invaded plots belonging to 42 families were identified in the three sampling sites of four major land use types. The number of families decreased by 23.8% in the infested area as compared to the non-invaded land unit. In invaded areas, of the 32 plant families, Poaceae accounts for 15.87%, followed by Fabaceae (14.29%) and Asteraceae (7.94%). Euphorbiaceae and Sapindaceae account for 4.76% each. The above-mentioned five families alone account for 47.62% of the plant species recorded in the Parthenium-invaded study areas.

Among 105 documented plant species, 63 species were common to both non-invaded and invaded areas. This means that species collected from non-*P. hysterophorus*-invaded sampling areas accommodate all the species that are found on the invaded plots. Subsequently, the result indicated that 98 indigenous, three endemic, and four introduced plant species were recorded from the entire species collection. Of these species, three of them were invasive (*Argemone mexicana*, *Parthenium hysterophorus*, and *Xanthium strumarium*).

### 3.1.2. Species Similarity and Richness

The index of similarity of species composition between areas of *P. hysterophorus* invaded and non-invaded land units within the same land use types for the four major selected land use types combined was high (Table 2).

**Table 2.** Similarity indices in *Parthenium* invaded and non-invaded areas.

| No. | Index of Similarity | Land Use Types | | | |
|-----|---------------------|------------------------|-----------------------------|------------------------|-------------------------|
|     |                     | Woodland Vegetation | Abandoned Agricultural Land | Roadside Vegetation | Grassland Vegetation |
| 1 | Jaccard's Similarity Index (IS$_j$) | 56.52 | 76.47 | 50 | 52.17 |
| 2 | Sorenson's Similarity Index (IS$_s$) | 72.22 | 86.66 | 66.66 | 68.57 |

Plant communities' similarity indices between *P. hysterophorus* non-invaded and invaded sites of the different land use types were >50% in all the study areas. The values of the Jaccard coefficient of similarity vary from 50% to 76.47%, whereas Sorensen varies from 66.6% to 86.6%. Similarity indices were comparatively higher on abandoned agricultural land, and a lower similarity index was recorded in the roadside vegetation (Table 2).

All of the 63 plant species found in the invaded area were found in the non-invaded area, and thus, species tend to be rarer in invaded areas. This was also evidenced by the higher frequencies for all of the species in non-invaded areas. At (=0.05), there is a significant difference in species richness between invaded and non-invaded areas. For each land use type, the effect of invasion on species richness was significant, i.e., the number of species varied significantly between *P. hysterophorus* invaded and non-invaded sites. The species richness also declined as *P. hysterophorus* densities increased (Figure 3).

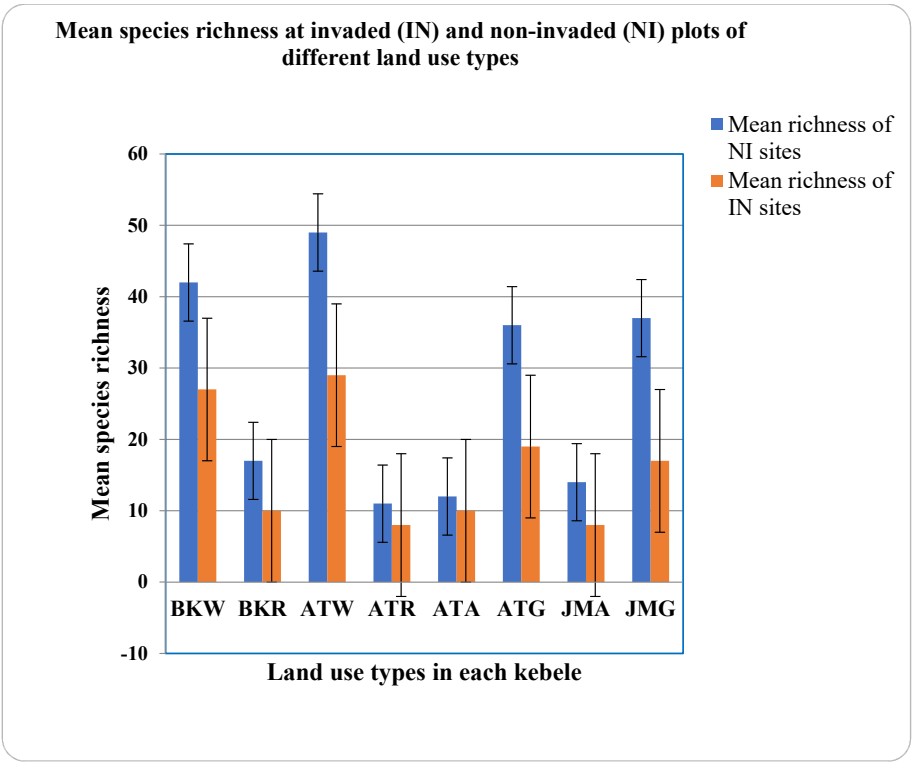

**Figure 3.** Comparison of species richness at invaded and non-invaded plots of different land use types. ATA—Ardatare Abandoned Agricultural land study side, ATW—Ardatare Woodland land study side, ATG—Ardatare Grazing land study side, ATR—Ardatare Roadside study side, JMA—Jamie Abandoned Agricultural land study side, JMG—Grazing land study side, BKR—Baliekecha Roadside study side, BKW—Baliekecha Woodland land study side.

### 3.1.3. Density, Percent Cover, and Height of *P. hysterophorus* across Different Land Use Types

The maximum mean density of *P. hysterophorus* stems was counted in the roadside area at 34 stems/m$^2$ and the minimum was in the woodland area at 14 stems/m$^2$. The mean density for an invaded area of grassland was 33 stem/m$^2$ and 282 stem/20 m$^2$ (14 stem/m$^2$) was recorded in a woodland forest area. Similarly, the mean density of *P. hysterophorus* for roadside and abandoned agricultural land use types was 34 and 32 stems/m$^2$, respectively (Table 3).

**Table 3.** Mean of *P. hysterophorus* density, height, and percentage cover in invaded and non-invaded land unit in Ginir district, southeastern Ethiopia.

| S.N | LUT | Study Site | Density (Stem/m$^2$) | | | Height (m) | | % Cover | |
| --- | --- | --- | --- | --- | --- | --- | --- | --- | --- |
| | | | NI | IN | NI | IN | NI | IN | |
| 1 | Acacia Wood land | Baliekecha | - | 15.825 | - | 0.53 | - | 60 | |
| | | Ardatare | - | 12.335 | - | 0.535 | - | 53 | |
| | | Mean | | 14.08 ± 1.745 | | 0.5325 ± 0.0025 | | 56.5 ± 3.5 | |
| 2 | Roadside | Baliekecha | - | 34.9 | - | 0.695 | - | 69 | |
| | | Ardatare | - | 33.5 | - | 0.565 | - | 75 | |
| | | Mean | | 34.2 ± 0.7 | | 0.63 ± 0.065 | | 72 ± 3.0 | |
| 3 | A. agri. land | Ardatare | - | 34 | - | 0.525 | - | 63 | |
| | | Jamie | - | 30 | - | 0.685 | - | 69 | |
| | | Mean | | 32 ± 2.0 | | 0.605 ± 0.08 | | 66 ± 3.0 | |
| 4 | G. land | Ardatare | - | 27.3 | - | 0.65 | - | 72 | |
| | | Jamie | - | 39.6 | - | 0.54 | - | 64 | |
| | | Mean | | 33.45 ± 6.15 | - | 0.595 ± 0.055 | - | 68 ± 4.0 | |

The highest density of *P. hysterophorus* was 61 stems/m$^2$ in the invaded area of grassland at Jamie's study site, and 47 stems/m$^2$ in the invaded area of abandoned agricultural land at Ardatare's study site. In the same way, 52 stems/m$^2$ was the maximum density ofX *P. hysterophorus* on the roadside of the Ardatare study site, and 365 stems/20 m$^2$ (18 stems/m$^2$) was for the woodland-invaded area of the Baliekecha sampling site.

The maximum mean of *P. hysterophorus* percent cover was 72% in the roadside area and, inversely, the minimum was 56.5% in the woodland area. In roadside study sites, the mean maximum percent cover of *P. hysterophorus* was 72%. In contrast, the minimum *Parthenium* percent cover was 56.5% in the Acacia woodland study sites (Table 3). The highest percent cover of *P. hysterophorus* (90%) was recorded on the Ardatare roadside, while the lowest percent cover (35%) was recorded at the woodland area of the Baliekecha sampling site. Consequently, the effect of land use types on the percent cover of *P. hysterophorus* was not significant at $p = 0.05$ (0.082).

The maximum mean height of *P. hysterophorus* was recorded at 0.63 m in the roadside area and the minimum mean height was 0.5325 m in the woodland area. The mean height of the invasive in woodland and grassland was 0.53 and 0.59 m, respectively. On the other hand, the mean height for the roadside and abandoned agricultural land in the invaded area was 0.63 and 0.60 m, respectively (Table 3). Individual heights of *P. hysterophorus* in invaded areas of woodland, grassland, roadside, and abandoned agricultural land revealed no significant differences in the invasive's mean height between these four different land use types at $p = 0.05$ (0.510).

### 3.1.4. Relationship between the Density of *P. hysterophorus* and Species Richness and Abundance

A scatter plot suggested that an increase in the density of *P. hysterophorus* led to a decrease in species abundance and the richness of native flora (Figure 4). The Pearson correlation ($r = -0.88$, $p = 0.001$) indicated a strong negative relationship between the density of *P. hysterophorus* and the species abundance in the invaded community. Sites that recorded a higher density of *P. hysterophorus* had lower floristic abundances of herbaceous and woody plant species. Similarly, sites that had a lower density of the invasive were recorded to harbor a higher species abundance. The decline in species richness and abundance with a

successive increase in invasive species density indicates that community heterogeneity and distribution are negatively affected significantly.

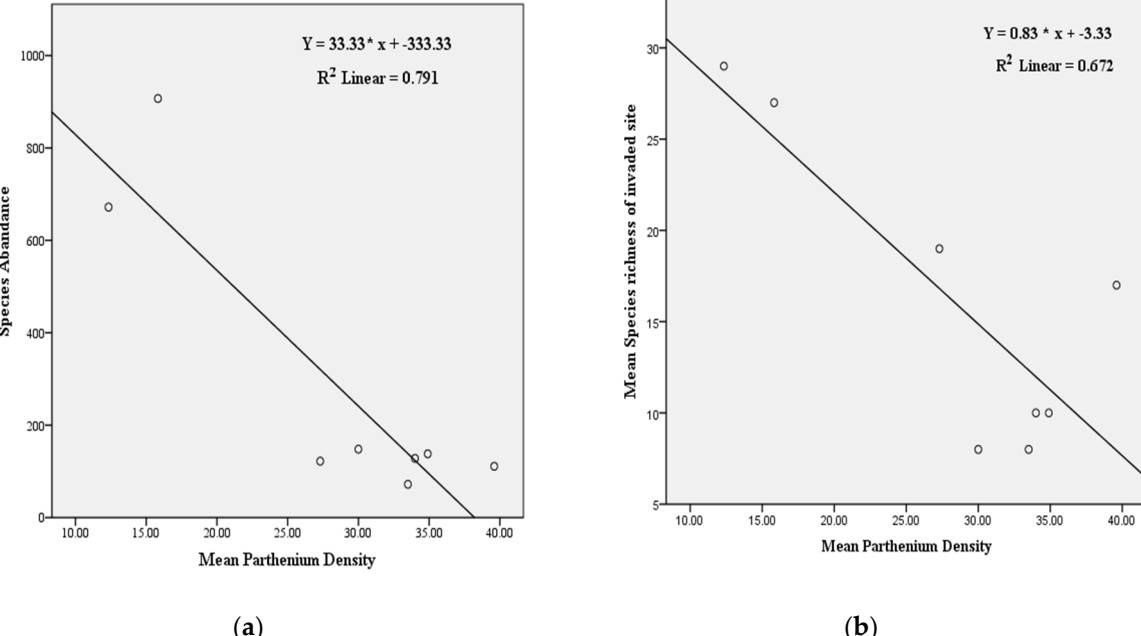

**Figure 4.** The relationship between mean *P. hysterophorus* density with species abundance and richness. (**a**). Mean *P. hysterophorus* density vs. mean species abundance. (**b**). Mean *P. hysterophorus* density vs. mean richness.

The regression analysis showed a strong negative relationship between the density of *P. hysterophorus* and the species richness per study site (Figure 4). The Pearson correlation also indicated that there was a significant negative relationship between the density of *Parthenium* and species richness (r = −0.82, *p* = 0.013) at *p* = 0.05. This implies that an increase in the density of the invasive led to a decrease in the number of other herbaceous and woody plant species in the sampling areas, hence the negative correlation values. Sites that had a low density of *P. hysterophorus* had more herbaceous and woody plant species.

## 4. Discussion

### 4.1. Impacts of P. hysterophorus on Indigenous Plant Biodiversity

*P. hysterophorus* had an impact on local biodiversity in the study area's various land use types, primarily in abandoned agricultural areas, grassland, roadsides, and woodland forest areas. In these land use types of the study sites, one can easily observe the prominent influence and fast expansion of *P. hysterophorus* on the other plant species. This may be due to many factors, such as wider adaptation across climates, photo insensitivity, and drought tolerance. The results of this study are consistent with [50,51]. Similarly, [52] described the allelopathic nature of the invasive and its impact on plant diversity.

According to the analysis, *P. hysterophorus* is the dominant species compared to other species in *P. hysterophorus*-infested areas. Very light or sometimes, no other vegetation can be seen in *P. hysterophorus*-dominated areas. On the contrary, better plant species diversity was recorded in non-infested areas. This might be due to the inhibitory nature of this invasive plant species. The result is in line with [53,54], who reported the inhibitory effect of allelochemicals of the invasive on both the germination and growth of a wide variety of crops, including pasture grasses, cereals, vegetables, and tree species.

Circumstantial evidence from the study indicated that *P. hysterophorus* has a great impact on plant diversity, causing habitat change in grasslands, along roadsides, open woodlands, and abandoned agricultural lands. These observations are in line with [55]. Ref. [56] also reported that *P. hysterophorus* can invade and adapt to new habitats, thereby

reducing the number of indigenous plants. A study by [57] also showed that *P. hysterophorus* is among one of the major invaders in the northwestern part of India, causing a huge loss to indigenous species diversity.

*Parthenium hysterophorus* does have a high negative impact on species richness in infested areas. In *P. hysterophorus* infested areas, it reduced the species diversity of the study area by reducing their distribution, abundance, and changing the ecosystem. These happen because when areas are invaded by *P. hysterophorus,* it reduces the growth and abundance of the different species, leading to the area being dominated by this invasive species. Similarly, [58] reported that the weed is among three exotic invasive species that adversely affect the structural composition and dynamics of the diversity of the native flora.

The decline in species diversity in *P. hysterophorus*-infested areas indicates that this invasive species is displacing certain native species from their community. In this study, higher diversity indices were relatively recorded in the non-invaded plots in all three study sites when compared to the *P. hysterophorus*-invaded plots. This is because *P. hysterophorus* alters the invaded ecosystem and species composition to such an extent that it threatens native flora. This response to invasion is to be predicted from earlier work in India [59].

The displacement of native species might also be related to their highly adaptive nature. It can flower under a very wide range of climatic conditions, creating a deep taproot that enables it to survive in low moisture, germinating at temperatures ranging from 12 °C to 27 °C, and tolerating saline conditions [60]. Therefore, the ecologically diversified adaptability of the invasive may allow its rapid expansion and speedup of damaging impacts on native plants, resulting in monoculture formation and native biodiversity reduction [54].

The variation in Shannon–Weiner species diversity indices and richness values in the three selected study sites in the invaded and non-invaded plots showed significant differences. The result of this study revealed that there was a reduction in the diversity index as the dominance of the weed increased. This finding was consistent with [61], who reported that there was a decline in the diversity index as the density of *P. hysterophorus* increased.

The result of this study was in agreement with [58], who reported the fact that plant diversity in the un-infested area was greater than in the *P. hysterophorus*-infested areas. Accordingly, a lower value of the Shannon diversity index suggests an area is dominated by a few species, i.e., invaded sites were relatively homogeneous in the community. Therefore, in the non-invaded study sites, the communities were more variable in the type of species and heterogeneous in the community. [62] also pointed out that the diversity values depend on the dominance of the weed in the community. If there are more successful species with no species completely dominating the area, the value of the Shannon diversity index is high.

There are a couple of possible reasons for the significant reduction in plant species that were present in highly infested areas. The dominance of *Parthenium* at the study site might be attributed to features such as its fast growth rate, which can grow and flower year-round without a period of dormancy; its high seed production, where individual plants can produce over 150,000 seeds in their lifetime, though the majority of plants produced less than 4000 seeds [63]; its adaptive nature, i.e., in unfavorable conditions it exhibits phenotypic plasticity, forming low-growing rosettes that only bolt and flower when conditions improve [60]; as well as its ability to remain viable in the soil for more than two years [27]. This concept is in line with the finding of [64], which described how, due to its high growth rate and short life cycle, Parthenium can quickly colonize sites, leading to its strong dominance in the habitats.

### 4.2. Impacts of P. hysterophorus on Species Similarity and Composition

The similarity index determines the interspecific association between the species of plant communities. The result of similarity indices between non-invaded and invaded areas of the different land use types in study areas was found to be high. This indicates that there is no fundamental change in species composition within the study sites. In agreement with this finding, [61] observed no major difference in species composition between infested and non-infested areas in the cultivated fields of Bilaspur, India.

In *P. hysterophorus*-invaded areas, species composition was found to be low compared to the non-invaded areas. In the studied area, the largest species proportion, which is 105 species (45 shrub/tree and 60 herbaceous) belonging to 42 families, was in the non-infested area, whereas 63 species belonging to 32 families were in the invaded area. The observed effects of *P. hysterophorus* on vegetation composition could be explained by the fact that it grows fast and spreads easily, thereby affecting the availability of resources. This is in line with the study by [26], which stated that the infestation of *P. hysterophorus* in the invaded area seriously distresses the composition and structure of plant species. In addition, [37] in a study conducted in the western part of Nyando Sub-County, Kenya, also postulated that an increase in *P. hysterophorus* would lead to changes in the structure and species composition of vegetation, affecting the availability of resources.

The identified families were also reported to be economically important and common in different parts of Ethiopia [65]. The most dominant plant species families, in terms of the number of species they contain, recorded in this research were analogous to other related findings, namely Fabaceae [30], Poaceae, Asteraceae [30,31,66] and Euphorbiaceae [31].

### 4.3. Impacts of P. hysterophorus Density on Species Richness and Abundance

The result of this study showed a negative relationship between the density of *Parthenium* and the density of native plant species in all selected land use types. Thus, the number of herbaceous, shrub, and tree species was found to be higher in areas where *P. hysterophorus* invasion was absent. [58] reported a decrease in species richness from 25 to 12 from a non-invaded Parthenium site to a heavily invaded Lower Himalaya (India) site. This would imply that most of the native species are not equipped with the adaptable characteristics of *P. hysterophorus* and cannot withstand its strong competition. These results support the work of [67], who reported a decline in different species due to an increase in *P. hysterophorus* density.

The density of *P. hysterophorus* in invaded plots found in the study area ranges from 12 to 61 stems/m$^2$ in the woodland area of the Ardatare site and grazing land of the Jamie study site, respectively. A study conducted by [36], in two urban areas of Nepal showed that the range of density of *P. hysterophorus* in invaded plots found in the Bharatpur area was 19 to 69 stems/m$^2$, which was similar to this study. The finding of this study is also comparable to the range reported by [68], which was 11 to 47 stems/m$^2$. Moreover, [28] reported 55 stems/m$^2$ in eastern Ethiopia, which was within the range of this study.

In contrast to the above reports, the [36] study at the Nepal Hetauda site found 402 stems/m$^2$, which was the highest record in Nepal and might also be the highest in the world. The highest stem *P. hysterophorus* density at the Hetauda site ecologically indicates that this weed acquires the most favorable habitat for its growth and germination. On the other hand, [69] reported lower densities, i.e., 1.5 to 38 stems/m$^2$ of the invasive in the grasslands of the mid hills in central Nepal. Furthermore, the *P. hysterophorus* density reported by [70] was 0.55 stems/m$^2$ in fallow lands, which was very low compared to the results of other studies, including this study.

The mean height of *P. hysterophorus* in invaded plots found in different land use types ranges between 53.25 cm and 63.0 cm, respectively. In this study, the lowest mean height was recorded in the woodland area, whereas the maximum mean height was recorded along the roadside. A study conducted by [31] in the Gedeo Zone, SNNPR (Southern Nations, Nationalities and Peoples Region), showed the highest mean height (1.6 m) of the invasive species along the way, which was not equivalent to this study. Another study by [71], indicated that the maximum height of *Parthenium* was in abandoned agricultural land. According to these authors and similar to this study, the *P. hysterophorus* height differed significantly ($p < 0.001$) with land use types

Several studies have revealed the aggressiveness of *P. hysterophorus* on native species [16,30,31,72]. A similar trend was observed in the current study where species diversity, richness, and density decreased significantly with the increase in *P. hysterophorus* density. For instance, sites that recorded a higher density of invasive species had the lowest

plant species richness and abundance. This makes *P. hysterophorus* a successful invader of non-native habitats [73].

During the initial stage of invasion, a very low density could be possible, but it quickly rises within a short time, and at this time, it may allow other species to grow in its vicinity. This is in agreement with the idea of [64], which reported that *P. hysterophorus* invasion at an early stage might increase habitat heterogeneity and grazing exclusion in grassland, roadside, plantation, and abandoned agricultural land. Overtime, as *P. hysterophorus* density increases, the richness of other species may decline. This may be because of the competitive replacement and lack of natural enemies outside its native range [73].

The decline in species diversity and richness with a continual increase in the *P. hysterophorus* infestation level is an indicator that the community heterogeneity has significantly and negatively been affected. This might be related to its ability to disperse animals and other human activities. The current results are in agreement with the findings of [30], which noted that within a few years of the introduction of the invasive into Awash National Park, there was a decline of 69% in the stand density of herbaceous species. Similarly, [51] also reported that the weed easily occupied new locations and often substituted native plant species, resulting in serious damage to biodiversity.

The mechanism of the decrease in species richness was explained by [74]. During the early stages of growth, *P. hysterophorus* forms a basal rosette of leaves that spreads rapidly very close to the ground and thus requires a suitable open area to establish. The stem of *P. hysterophorus* then elongates rapidly and starts branching at the apex, and this interferes with the emergence of other seedlings. Moreover, due to its high growth rate, *P. hysterophorus* becomes competitive and develops the ability to exclude the growth of other species.

In agreement with the above concept, [75] describe how native species differ in their resistance to invasion, i.e., some are excluded from invaded communities more easily than others are. In this study, *Argemone mexicana*, *Calpurina aurea*, *Cynodon dactylon*, *Datura stramonium*, *Dodonaea angustifolia*, *Euclea racemosa*, *Solanum incanum*, and *Xanthium strumarium* were the major and relatively dominant species as they were present in most of the selected land use types. This might be because the plants have strong competitive vigor with *P. hysterophorus* and are also adaptable to the different land use types.

## 5. Conclusions

The results of the current study have indicated that *P. hysterophorus* harms indigenous species composition by decreasing species diversity. The study demonstrated that infestation of the invasive was found to have a highly negative impact on the species richness and abundance of herbaceous and woody species of vegetation by reducing their growth and distribution and by changing the habitat of an area. In this study, higher diversity indices were relatively recorded in the non-invaded areas in all three study sites when compared to invaded areas. This study also discovered that *P. hysterophorus* infestation changes the structure of plant species in the invaded community, as the mean density of *P. hysterophorus* was found to be significantly different across land use types. The adverse effect of the invasive was mainly notable on plant species of grassland, woodland and bushland, roadside, and abandoned agricultural land areas where these land use types are the major feed sources for livestock and are areas of high plant diversity. This in turn reduced the carrying capacity of grazing lands and the diversity of plants that existed in the study area.

Based on the results obtained, the following recommendations were made: creating public awareness through different means of communication about its impacts on plant species diversity for the local community and relevant stakeholders to prevent its further spread into agriculture lands and other natural ecosystems of the study area; the priority kebeles in the district could be identified based on the density distribution of the invasive to act accordingly and to control its further dissemination; protecting and restoring those identified as sensitive and important land use types through an integrated multidisciplinary

approach; further wider and long-term research could be conducted on its impacts on crop yield, human and domestic animals' health, biodiversity of indigenous plant species, and soil seed bank; removing this invasive species through a public campaign prior to seed production, and the Ethiopian Biodiversity Institute could take responsibility for an integrated long-term management program by coordinating local people, universities, research centers, governments, and non-governmental organizations to work together.

**Author Contributions:** Conceptualization, M.B.; methodology, M.B.; software, M.B.; validation, M.B., Z.G. and G.D.; formal analysis, M.B.; investigation, M.B.; resources, M.B., Z.G. and G.D.; data curation, M.B.; writing—original draft preparation, M.B.; writing—review and editing, Z.G. and G.D.; visualization, M.B.; supervision, Z.G. and G.D.; project administration, M.B.; funding acquisition, M.B. and Z.G. All authors have read and agreed to the published version of the manuscript.

**Funding:** This research received no external funding.

**Data Availability Statement:** Not applicable.

**Acknowledgments:** Our sincere gratitude goes to the Ethiopian Biodiversity Institute for its financial support to carry out this research and for the provision of field materials and plant identification. We are also highly grateful to Ginir district environment, forest and climate change authority and district agricultural offices for their provision of information and support letters during data collection.

**Conflicts of Interest:** The authors declare no conflict of interest.



### Appendix A. Number of Each Species in Each Land Use Types

| NO. | Botanical Name | Local Name | | Family Name | Road side | Grazing Land | Acacia Woodland | Abandoned Agri. Land | No. of Individuals |
| | | Oromic | Amharic | | | | | | |
|---|---|---|---|---|---|---|---|---|---|
| 1. | *Acacia albida* Del. | Garbi | | Fabaceae | 0 | 3 | 9 | 0 | 12 |
| 2. | *Acacia brevispica* Harms | Hamaresa | Qwanta | Fabaceae | 6 | 4 | 12 | 0 | 22 |
| 3. | *Acacia bussei* Harms ex sjostede | Halo | | Fabaceae | 0 | 0 | 3 | 0 | 3 |
| 4. | *Acacia etbaica* Schweinf. | Derie; Qereta | | Fabaceae | 0 | 2 | 5 | 0 | 7 |
| 5. | *Acacia gerrardii* Benth. | Dodoti | | Fabaceae | 0 | 2 | 4 | 0 | 6 |
| 6. | *Acacia mellifera* (Vahl.) Benth. | Bilala; | Kontir; Atnkuy | Fabaceae | 0 | 3 | 7 | 0 | 10 |
| 7. | *Acacia nilotica* (L.) Willd. ex. Del. | Burquqe; Kasale | Burquqe | Fabaceae | 0 | 0 | 3 | 0 | 3 |
| 8. | *Acacia oerfota* (Forssk.) Schweinf. | Wangay | | Fabaceae | 0 | 0 | 6 | 0 | 6 |
| 9. | *Acacia senegal* (L.) Willd. | Gorsa | Kontir | Fabaceae | 0 | 2 | 5 | 0 | 7 |
| 10. | *Acacia seyal* Del. | Wacho | | Fabaceae | 0 | 2 | 7 | 0 | 9 |
| 11. | *Acacia tortilis* (Forssk.) Hayne. | Tadacha; Korera | | Fabaceae | 0 | 1 | 6 | 0 | 7 |
| 12. | *Acalypha racemosa* Baill. | Dhigri | | Euphorbiaceae | 0 | 2 | 5 | 0 | 7 |
| 13. | *Acokanthera schimperi* (A.DC) Schwein. | Qararu | Merenz | Apocynaceae | 0 | 0 | 4 | 0 | 4 |
| 14. | *Agave sisalana* Perro ex Eng. | Alge; Qaca | | Agavaceae | 2 | 6 | 5 | 0 | 13 |
| 15. | *Aloe pirottae* Berger. | Hargisa baru | | Aloaceae | 2 | 0 | 6 | 0 | 8 |
| 16. | *Aloe retrospiciense* Reynolds and Bally | Hargisa | Ret | Aloaceae | 3 | 0 | 7 | 0 | 10 |
| 17. | *Andropogon gayanus* Kunth. | Gaja | | Poaceae | 0 | 274 | 0 | 0 | 274 |
| 18. | *Argemone mexicana* L. | Qore adi | Nech Lebash | Papaveraceae | 36 | 23 | 11 | 52 | 122 |
| 19. | *Asparagus falcatus* L. | Seriti | Yeseit qest | Asparagaceae | 3 | 2 | 8 | 0 | 13 |
| 20. | *Avena abyssinica* Hochst. | Gaja gaca | Sinar | Poaceae | 0 | 0 | 0 | 23 | 23 |
| 21. | *Balanites aegyptiaca* (L.) Del. | Bedena | Jemo; Kudkuda | Balanitaceae | 0 | 0 | 4 | 0 | 4 |
| 22. | *Barbeya oleoides* Schweinf. | Adado | | Barbeyaceae | 0 | 0 | 12 | 0 | 12 |
| 23. | *Barleria eranthemoides* R. Br. | Balanwaranti | YesetAfe | Acanthaceae | 0 | 14 | 3 | 0 | 17 |
| 24. | *Bothriochloa insculpta* (Hochst. ex A.Rich.) | Suto | | Poaceae | 0 | 286 | 0 | 0 | 286 |
| 25. | *Cadia purpurea* (Picc.) Ait. | Tokeda; Hijire | | Fabaceae | 2 | 0 | 6 | 0 | 8 |
| 26. | *Calpurina aurea (Ait.) Benth.* | Cheketa | Digta | Fabaceae | 14 | 7 | 35 | 0 | 56 |
| 27. | *Canthium pseudosetitflorum* Bridson | Ladhana | | Rubiaceae | 0 | 0 | 3 | 0 | 3 |
| 28. | *Caralluma speciosa* (N. E.Br.) | Haleko aje | | Asclpiadaceae | 0 | 0 | 4 | 0 | 4 |
| 29. | *Carissa spinarum* L. | Hagamsa | Agam | Apocynaceae | 2 | 3 | 5 | 0 | 10 |
| 30. | *Cissus cactiformis* Gilg. | Gorsa; Matbot | Guraj | Vitaceae | 0 | 0 | 6 | 0 | 6 |
| 31. | *Cissus quadrangularis* L. | Chophi | | Vitaceae | 0 | 4 | 3 | 0 | 7 |

Appendix A. *Cont.*

| NO. | Botanical Name | Local Name | | Family Name | Road side | Grazing Land | Acacia Woodland | Abandoned Agri. Land | No. of Individuals |
|-----|----------------|------------|---|-------------|-----------|--------------|-----------------|----------------------|--------------------|
| | | Oromic | Amharic | | | | | | |
| 32. | *Clematis sp.* | Gadila | | Ranunculaceae | 22 | 6 | 0 | 0 | 28 |
| 33. | *Combretum molle* R. Br. ex G. Don | Biresa; | Abalo; Weyba | Combertaceae | 0 | 0 | 7 | 0 | 7 |
| 34. | *Commiphora africana* (A. Rich) Engl. | Hammesa | Anquwa | Burseraceae | 0 | 0 | 5 | 0 | 5 |
| 35. | *Commiphora confusa* Vollesen | Chacho; Hamesa | | Burseraceae | 0 | 0 | 2 | 0 | 2 |
| 36. | *Commiphora erythraea* (Ehrenb.) Engl. | Hagarsu | | Bursseraceae | 0 | 0 | 11 | 0 | 11 |
| 37. | *Commiphora sp.* | Dhiga | | Bursseraceae | 0 | 0 | 2 | 0 | 2 |
| 38. | *Conomitra linearis* Fenzl | Hanchagire | | Asclepiadaceae | 0 | 0 | 9 | 0 | 9 |
| 39. | *Cordia africana* Lam. | Wadesda | Wanza | Boraginaceae | 0 | 0 | 5 | 0 | 5 |
| 40. | *Crepis rueppellii* Sch. Bip. | Aanano | Yefyel Wotet | Asteraceae | 0 | 5 | 0 | 9 | 14 |
| 41. | *Croton macrostachyus* Del. | Bakanisa | Bissana | Euphorbiaceae | 3 | 4 | 6 | 0 | 13 |
| 42. | *Cucumis prophetarum* L. | | Yemdere embway | Cucurbitaceae | 0 | 2 | 0 | 0 | 2 |
| 43. | *Cymbopogon commutatus* (Steud.) Stapf | | Sembelet | Poaceae | 0 | 258 | 0 | 0 | 258 |
| 44. | *Cynodon dactylon* (L.) Pers. | Sardo | | Poaceae | 0 | 315 | 0 | 18 | 333 |
| 45. | *Cynodon aethiopicus* Clayton and Harlan | | | Poaceae | 0 | 276 | 0 | 0 | 276 |
| 46. | *Cynoglossum geometricum* Hochst. ex A.DC. | Matane-chati | Chigogot | Boraginaceae | 8 | 0 | 0 | 26 | 34 |
| 47. | *Datura stramonium* L. | Banji | Atse Faris | Solanaceae | 46 | 0 | 0 | 23 | 69 |
| 48. | *Dichrostachys cinerea* L. | Adesa; Jirme | Ader | Fabaceae | 0 | 0 | 6 | 0 | 6 |
| 49. | *Digitaria abyssinica* (Hochst ex. A.Rich.) Stapf | Wariat | | Poaceae | 0 | 252 | 0 | 0 | 252 |
| 50. | *Dodonaea angustifolia* L.f. | Etecha | Kitkita | Sapindaceae | 30 | 12 | 48 | 0 | 90 |
| 51. | *Dombeya torrida* (J. F. Gmel.) P. Bamps | Danisa | Wulkfa | Sterculiaceae | 0 | 0 | 5 | 0 | 5 |
| 52. | *Dracaena ellenbeckiana* Engler | Metti; Yabelo | | Deracenaceae | 0 | 0 | 3 | 0 | 3 |
| 53. | *Ehretia cymosa* Thonn. | Ulaga; Mukereba | | Boraginaceae | 0 | 0 | 2 | 0 | 2 |
| 54. | *Eleusine floccifolia* (Forssk.) Spreng. | Dagoo | Akirma | Poaceae | 0 | 216 | 0 | 0 | 216 |
| 55. | *Enteropogon macrostachyus* (Hochst ex A.Rich.) Benth. | | | Poaceae | 0 | 235 | 0 | 0 | 235 |
| 56. | *Eragrostis papposa* (Roem. and Schult.) Steud. | | | Poaceae | 0 | 292 | 0 | 0 | 292 |
| 57. | *Erythrina brucei* Schweinf. | Walena | Korch | Fabaceae | 0 | 0 | 1 | 0 | 1 |
| 58. | *Euclea racemosa* subsp. *schimperi* | Mieasa | Dedeho | Ebenaceae | 12 | 24 | 62 | 0 | 98 |
| 59. | *Euphorbia dumalis* S. Carter | Guri | Anterfa | Euphorbiaceae | 0 | 0 | 6 | 0 | 6 |
| 60. | *Euphorbia sp.* | | Qulqwalit | Euphorbiaceae | 0 | 6 | 0 | 0 | 6 |
| 61. | *Euphorbia tirucalli* L. | | Kinchib | Euphorbiaceae | 2 | 0 | 6 | 0 | 8 |

Appendix A. *Cont.*

| NO. | Botanical Name | Local Name | | Family Name | Road side | Grazing Land | Acacia Woodland | Abandoned Agri. Land | No. of Individuals |
|-----|----------------|------------|---|-------------|-----------|--------------|-----------------|---------------------|--------------------|
| | | Oromic | Amharic | | | | | | |
| 62. | *Ficus sycomorus* L. | Oda | | Moraceae | 0 | 0 | 2 | 0 | 2 |
| 63. | *Ficus vasta* Forssk. | Qiltu | Warka | Moraceae | 0 | 0 | 3 | 0 | 3 |
| 64. | *Grewia mollis* A. Juss. | Haroresa | Betre Musie | Tiliaceae | 0 | 0 | 4 | 0 | 4 |
| 65. | *Guizotia scabra* (Vis.) Chiov. | Hadaa; Tufo | Gime | Asteraceae | 0 | 0 | 0 | 42 | 42 |
| 66. | *Guizotia schimperi* Sch. Bip. ex Walp. | Hadaa; | | Asteraceae | 0 | 0 | 0 | 34 | 34 |
| 67. | *Haplocoelum foliolosum* (Hiem) Bullock | Chena | Adey Abeba | Sapindaceae | 0 | 0 | 11 | 0 | 11 |
| 68. | *Hibiscus macranthus* Hochst. ex A. Rich | Sukumeta; | Nacha | Malvaceae | 4 | 0 | 0 | 0 | 4 |
| 69. | *Hyparrhenia hirta* (L.) Stapf | | Sembeliet | Poaceae | 0 | 296 | 0 | 0 | 296 |
| 70. | *Ipomoea kituiensis* Vatke | Gale; kossole | | Convolvulaceae | 0 | 0 | 5 | 0 | 5 |
| 71. | *Juniperus procera* Hochst. Ex Endl. | Hindhesa | Tid | Cupressaceae | 0 | 0 | 10 | 0 | 10 |
| 72. | *Justicia schimperiana* (Hochst. ex Nees) T. Anders. | Dumoga | Sensel | Acanthaceae | 14 | 0 | 6 | 0 | 20 |
| 73. | *Kalanchoe petitiana* A.Rich. | Hancura | | Crassulaceae | 5 | 0 | 7 | 0 | 12 |
| 74. | *Lactuca inermis* Forssk. | Mech Algu | | Asteraceae | 3 | 4 | 0 | 9 | 16 |
| 75. | *Lannea schimperi* (A. Rich) Engl. | Ruku; Rukesa | | Anacardiaceae | 0 | 0 | 4 | 0 | 4 |
| 76. | *Launaea intybacea* (Jacq.) Beauv. | Hoola-gabbisa, | | Asteraceae | 0 | 8 | 0 | 26 | 34 |
| 77. | *Leucas martinicensis* (Jacq.) R. Br. | | Yeferes Zeng | Lamiaceae | 0 | 3 | 0 | 17 | 20 |
| 78. | *Lippia adoensis* Hochst. ex Walp. | Kassie | Kassie | Verbenaceae | 0 | 6 | 0 | 0 | 6 |
| 79. | *Maerua aethiopica* (Fenzl) Oliv. | | Kontr | Capparidaceae | 0 | 3 | 0 | 0 | 3 |
| 80. | *Olea europaea* L. subsp. cuspidata | Ejersa | Wieyra | Oleaceae | 0 | 2 | 10 | 0 | 12 |
| 81. | *Opuntia ficus-indica* (L.) Miller. | Shonka | Qulqual | Cactaceae | 16 | 0 | 24 | 0 | 40 |
| 82. | *Osteospermum vailliantii* (Decne) T. Norl. | Gurbi halooftu | | Asteraceae | 0 | 0 | 0 | 31 | 31 |
| 83. | *Ozoroa insignis* Del. | Garri | | Anacardiaceae | 0 | 0 | 3 | 0 | 3 |
| 84. | *Pappea capensis* Eckl. and Zeyh. | Biqa | | Sapindaceae | 0 | 0 | 3 | 0 | 3 |
| 85. | *Parthenium hysterophorus* L. | Anamale; Faramsis | | Asteraceae | | | | | |
| 86. | *Pennisetum sphacelatum* (Nees) Th. Dur. and Schinz | Geta | Sendedo | Poaceae | 0 | 244 | 0 | 0 | 244 |
| 87. | *Psydrax schimperiana* (A. Rich) Bridson | Galoo; Seged | | Rubiaceae | 0 | 0 | 8 | 0 | 8 |
| 88. | *Rhoicissus revoilii* Planch. | Aremo Saged | | Vitaceae | 0 | 0 | 5 | 0 | 5 |
| 89. | *Rhus natalensis* Krauss. | Gongoma | mst-aybelash | Anacardiaceae | 0 | 0 | 6 | 0 | 6 |
| 90. | *Rhus vulgaris* Meikle | Tatesa | Embs | Anacardiaceae | 0 | 0 | 4 | 0 | 4 |
| 91. | *Ricinus communis* L. | Kobo | Gulo | Euphorbiaceae | 38 | 0 | 0 | 0 | 38 |
| 92. | *Rosa abyssinica* Lindley | Gora | Kega | Rosaceae | 6 | 0 | 15 | 0 | 21 |

**Appendix A.** *Cont.*

| NO. | Botanical Name | Local Name | | Family Name | Road side | Grazing Land | Acacia Woodland | Abandoned Agri. Land | No. of Individuals |
|---|---|---|---|---|---|---|---|---|---|
| | | Oromic | Amharic | | | | | | |
| 93. | *Rumex nepalensis* Spreng. | | Tult | Polygonaceae | 0 | 24 | 0 | 46 | 70 |
| 94. | *Secamone Parvifolia* (Olive.) Bullock | sari | | Asclepiadaceae | 0 | 0 | 6 | 0 | 6 |
| 95. | *Senna didymobotrya* (Fresen.)Irwin and Barneby | | | Fabaceae | 28 | 0 | 0 | 0 | 28 |
| 96. | *Senra incana* Cav. | | Nechilo | Malvaceae | 0 | 5 | 10 | 0 | 15 |
| 97. | *Snowdenia polystachya* (Fresen.) Pilg. | Muja | | Poaceae | 0 | 32 | 0 | 36 | 68 |
| 98. | *Solanum incanum* L. | Hidi | Embwa'y | Solanaceae | 54 | 37 | 0 | 28 | 119 |
| 99. | *Sporobolus africanus* (Poir.) Robyns and Tournay | Murie; Migra | | Poaceae | 0 | 192 | 0 | 0 | 192 |
| 100. | *Tagetes minuta* L. | Aje | Gime;Yahiya Ariti | Asteraceae | 0 | 0 | 0 | 74 | 74 |
| 101. | *Tamarandus indica* L. | | Roqa | Fabaceae | 0 | 0 | 2 | 0 | 2 |
| 102. | *Terminalia brownii* Fresen. | Biresa; Weyba | | Combretaceae | 0 | 0 | 7 | 0 | 7 |
| 103. | *Woodfordia uniflora* (A. Rich) Koehne. | Dambitto; Mar Mate | | Lythraceae | 0 | 0 | 2 | 0 | 2 |
| 104. | *Xanthium strumarium* L. | Bandoo Abdulhakim | | Asteraceae | 17 | 9 | 0 | 23 | 49 |
| 105. | *Ziziphus mauritiana* Lam. | Kurkura | | Rhamnaceae | 0 | 0 | 4 | 0 | 4 |
| | | | | | 378 | 3408 | 541 | 517 | 4844 |

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
