# Peer review of "Impacts of Parthenium hysterophorus L. on Plant Species Diversity in Ginir District, Southeastern Ethiopia"

_diversity, doi:10.3390/d14080675_

Round 1
Reviewer 1 Report
The manuscript presented for review, entitled “Impacts of Parthenium hysterophorus Linn on Herbaceous and Woody Species Diversity and Community Structures ………..” concerns the important issue of expansion of alien species causing significant ecological and economical losses. Understanding the influence of these species on the local flora and underlying mechanisms may be of great environmental importance, hence this type of study could play an important role.
The presented study, although it concerns such an important issue, requires major corrections before it will be accepted for publication. The amendments concern the content-related and linguistic part, as well as the citation of literature and the preparation of figures.
1. First: I will propose to change the title: Impact of Parthenium hysterophorous L. on species diversity in….
2. I will put in keywords: alien species, weed, Santa Maria feverfew
3. In the introduction section, a general description of the species Parthenium is necessary: ​​its natural and secondary distribution, taxonomical and biological description, history of invasion, examples of environmental and economic costs caused by this species, etc. Not all readers are familiar with this weed.
4. Materials and methods: In general, the numbering of subsections in the manuscript is inconsistent
Subsection 2.2.3.2. Authors should combine subsections. There is no need for a detailed description of the collecting and drying of plants as this is a standard method. I propose to quote the source of Flora and the place where the specimens are stored. In the acknowledgments, the authors should also thank for the plants identifying
Subsection 2.2.5 (?, lack of 2.2.4.) Data analysis: the chapter should be significantly shortened. The applied species diversity coefficients are commonly known, there is no need to provide their formulas and discuss their interpretation in detail.
The authors describe in detail the Shannon and Evenness coefficients, but it is not known what is meant by the abundance (the total number of occurrences of specimens in the plots?). Significance tests for differences between data groups would also be useful.
5. Section 3. Results and discussion – this section should be divided into separate results and discussion. a chapter on Results and a separate Discussion should be separated. 3.1.1 - There is no ecological justification for the use and interpretation of the “overall mean” due to the different types of communities (text and table 1). Please note how is the range of the data for Abundance between roadside and grassland.
6. As the authors refer to the species richness, their numbers, and their families, it would be good to include the lists of species present in each type in the Appendix (e.g. a collective list in the table with the occurrence in the type of communities in the columns).
7. Subsection 3.1.2. the title of the subsection is inadequate to its content – the authors describe the density of Parthenium stems, not community structure. The term community structure should be removed from the results, as the authors do not literally analyze this issue as a separate one.
8. Subsection 4.1.2.1 (incorrect numbering) - the title should be changed and the height of Parthenium removed - the results concerning the Parthenium height are (and should be) combined with information on the weed density.
9. 3.2 The discussion should be rearranged so as to connect the threads to be discussed, e.g. the authors refer to allelopathy several times in different places: lines; 375, 381, 409, 496. They should discuss that problem ones but thoroughly. Similarly, they should refer to plant diversity. In the discussion, the authors do not refer to the knowledge of the species biology and causes of its expansion They only write in general terms: fast growth rate, high seed production, adaptative nature (line. What is adaptative nature of this species? It would also be interesting if the authors consulted the papers on how in some countries attempts are made to limit the incidence of this nuisance weed. You can benefit from the DOI report: 10.1111 / epp.12168.
Figures:
Fig. 1: the lower map is stretched
Fig. 2: rearrange the description – first Fig. 2. ….., later notes…
In Fig 2 the plots should be quadrats (not rectangles)
References and citations:
The authors use names and last names both in the text (e.g. l. 62, 64, 65x2 etc.) and in the references (e.g. l. 587, 591, 594 etc.). Sometimes they shorted last names (e.g. l. 110).
Even, at this time of review, where there are no rules for the references style, the authors should be consequent in the style used.There is a big disorder in the references.
There are also a lack of some cited studies e.g. Ayele et al. 2014.
Style of the text:
there is a need for carefully linguistic preparation of the text. As English is not my mother tongue I will not give specific corrections.
Others:
Use plots not quadrats (e.g. in Abstract); be careful with space between P. and hysterophorous; use a.s.l.; add space before m (meter) and remove space before %; be aware of the italic when use Latin names; transcription of study sites names (Fig. 3); use alphabetic order when you list the species, etc.
Author Response
Response to Reviewer 1 Comments
Reviewer 1
Comments - 1: First: I will propose to change the title: Impact of Parthenium hysterophorous L. on species diversity in….
Response – 1: The title of the paper is modified as “Impacts of Parthenium hysterophorus L. on Plant Species Diversity in Ginir District, Southeastern Ethiopia”. We the authors also believe that in this way it is more descriptive containing the fewest possible words needed to adequately describe the content and/or purpose of our research paper.
Comments - 2: I will put in keywords: alien species, weed, Santa Maria feverfew
Response – 2: Keywords like alien species, weed, Santa Maria feverfew, composition, herbaceous and woody plants are also added according to their alphabetical order as they are also among the most important lists of key words and which can possibly reflect the contents of our paper.
Comments - 3: In the introduction section, a general description of the species Parthenium is necessary: ​​its natural and secondary distribution, taxonomical and biological description, history of invasion, examples of environmental and economic costs caused by this species, etc. Not all readers are familiar with this weed.
Response - 3: In the introduction section, as the first reviewer comments, some general descriptions on Parthenium i.e. its taxonomy and biology, history of invasion, its dispersal methods and estimated economic costs worldwide are included in its appropriate place so as to improve the introduction section.
Comments - 4: Materials and methods: In general, the numbering of subsections in the manuscript is inconsistent
Subsection 2.2.3.2. Authors should combine subsections. There is no need for a detailed description of the collecting and drying of plants as this is a standard method. I propose to quote the source of Flora and the place where the specimens are stored. In the acknowledgments, the authors should also thank for the plants identifying
Subsection 2.2.5 (?, lack of 2.2.4.) Data analysis: the chapter should be significantly shortened. The applied species diversity coefficients are commonly known, there is no need to provide their formulas and discuss their interpretation in detail.
The authors describe in detail the Shannon and Evenness coefficients, but it is not known what is meant by the abundance (the total number of occurrences of specimens in the plots?). Significance tests for differences between data groups would also be useful.
Response – 4: In Materials and methods, the numbering of subsections was due to typing error they are now corrected accordingly. As it was also commented subsections under sampling design and data collection are merged. Some detail descriptions like standard methods of collection are minimized. For Herbarium identification process and support of the Published Ethiopian and Eritrean Flora books the Ethiopian Biodiversity Institute is acknowledged.
In data analysis again it was tried to shorten the detailed descriptions on species diversity coefficients and the commonly known formulas and their detail discussion and interpretations was removed. The concept of abundance is also described so as to give insight for readers.
Comments - 5: Section 3. Results and discussion – this section should be divided into separate results and discussion. a chapter on Results and a separate Discussion should be separated. 3.1.1 - There is no ecological justification for the use and interpretation of the “overall mean” due to the different types of communities (text and table 1). Please note how the range of the data is for Abundance between roadside and grassland.
Response – 5: The section of Results and discussions are independently divided into separate results and discussion. The overall mean is also removed from tables and from the text since it have no ecological justification interpreting this concept as commented.
Comments - 6: As the authors refer to the species richness, their numbers, and their families, it would be good to include the lists of species present in each type in the Appendix (e.g. a collective list in the table with the occurrence in the type of communities in the columns).
Response - 6: As commented by the reviewer, all assessed species in each land use types with their occurrence are attached as appendix at the end of the paper.
Comments - 7: Subsection 3.1.2. the title of the subsection is inadequate to its content – the authors describe the density of Parthenium stems, not community structure. The term community structure should be removed from the results, as the authors do not literally analyze this issue as a separate one.
Response - 7: The subsection 3.1.2 as commented is removed and the title is modified as “Density, Percent cover and height of Parthenium across different land use types” under sub section 3.1.3. In the discussion section the concern of community structure is contained within the subsection 4.3.
Comments - 8: Subsection 4.1.2.1 (incorrect numbering) - the title should be changed and the height of Parthenium removed - the results concerning the Parthenium height are (and should be) combined with information on the weed density.
Response - 8: Including subsection 4.1.2.1 correction, the shifts on numbering have been done. The title under this subsection is also improved as “Relationship between density of Parthenium and species richness and abundance” and the result of Parthenium height is combined under subsection 3.1.2.
Comments - 9: 3.2 The discussion should be rearranged so as to connect the threads to be discussed, e.g. the authors refer to allelopathy several times in different places: lines; 375, 381, 409, 496. They should discuss that problem ones but thoroughly. Similarly, they should refer to plant diversity. In the discussion, the authors do not refer to the knowledge of the species biology and causes of its expansion. They only write in general terms: fast growth rate, high seed production, adaptive nature (line. What is adaptive nature of this species? It would also be interesting if the authors consulted the papers on how in some countries attempts are made to limit the incidence of this nuisance weed. You can benefit from the DOI report: 10.1111 / epp.12168.
Response - 9: The discussions are rearranged considering all the given comments. Some repetitive ideas are merged and discussed briefly adding some more descriptive ideas e.g. concept of allelopathy. The reference paper attached by the reviewer was also checked and some important concepts are taken from it. These all amended texts are colored to distinguish them from the former ones.
General comments
Comment 1: Figures
Fig. 1: the lower map is stretched
Fig. 2: rearrange the description – first Fig. 2. ….., later notes…
In Fig 2 the plots should be quadrats (not rectangles)
Responses 1: Regarding the figures, all are corrected as per the given comments. The stretched map is corrected. The description under figure 2 is rearranged and plots of the figures are readjusted as the layout is concerned to indicate quadrats.
Comment 2: References and citations
- The authors use names and last names both in the text (e.g. l. 62, 64, 65x2 etc.) and in the references (e.g. l. 587, 591, 594 etc.). Sometimes they shorted last names (e.g. l. 110).
- Even, at this time of review, where there are no rules for the references style, the authors should be consequent in the style used. There is a big disorder in the references.
- There are also a lack of some cited studies e.g. Ayele et al. 2014.
Responses 2: All comments given regarding references and citations are corrected in deep considering Harvard reference style. The missed references are also fulfilled and some other new citations are also added as per the rule of references mentioned above.
Comment 3: Style of the text
- There is a need for carefully linguistic preparation of the text. As English is not my mother tongue I will not give specific corrections.
Responses 3: Linguistic concerns were carefully reviewed all over the paper considering the comments given.
Comment 4: Others
- Use plotsnot quadrats (e.g. in Abstract); be careful with space between P. and hysterophorous; use a.s.l.; add space before m (meter) and remove space before %; be aware of the italic when use Latin names; transcription of study sites names (Fig. 3); use alphabetic order when you list the species, etc.
Responses 4: In all sections including the abstract the term quadrats are replaced by plots and a space between units, terms and characters are recognized as commented. Regarding the transcription, the names of the study sites are locally transcribed differently by different local residents so it is difficult to have common canned names. List of species of all over the paper are rearranged according to their alphabetic order.

Reviewer 2 Report
The design appears appropriate, however the inferential statistics are very thin, poorly described, and probably wrong for this type of research. What is meant by species abundance? This could be critical in interpretation, and I believe it's not what is intended to be. I don't see error bars or standard deviation values in the figures or tables. It's thus impossible to comment on how correct the conclusions are. The language is poor, and the presentation is wanting in many other ways.
More specific comments:
Author name is abbreviated L., not Linn.
Family names should not be italicised, but genus and species should, throughout.
"District' and other administrative units should not be capitalised.
Formulas must be types, not pasted as figures.
Some statements need to be qualified. Africa to be worst hit by climate change? In what respect, and is that relevant here?
Author Response
Response to Reviewer 2 Comments
Reviewer 2
- General Comments: The design appears appropriate, however the inferential statistics are very thin, poorly described, and probably wrong for this type of research. What is meant by species abundance? This could be critical in interpretation, and I believe it's not what is intended to be. I don't see error bars or standard deviation values in the figures or tables. It's thus impossible to comment on how correct the conclusions are. The language is poor, and the presentation is wanting in many other ways.
Responses: Species abundance is defined in its appropriate place as it is important for interpretation. I also tried to revise the statistical analysis, standard error for all mean values and standard error bar in tables and figure was contained within. A language and grammar issue was revised all over the paper
- More specific comments
Author name is abbreviated L., not Linn.
Family names should not be italicized, but genus and species should, throughout.
"District' and other administrative units should not be capitalized.
Formulas must be types, not pasted as figures.
Some statements need to be qualified. Africa to be worst hit by climate change? In what respect, and is that relevant here?
Responses: The name of the author is corrected as L., by removing Linn. and I leave italicizing family names of species while genus and species names throughout the paper are written in Italic. District and other administrative units that were previously capitalized are now written in lowercase. Copy and pasted formulas and detail descriptions are removed as they are commonly known and unnecessary as commented by the reviewers. In addition, some statements are arranged and their contexts are shaped as per the given comments. All of these are highlighted so as to identify clearly.

Round 2
Reviewer 1 Report
The authors of the manuscript made the significant effort to improve the quality of the paper. There are still some things to change according to previous review. I found the Introduction with the description the species in general fine. Also, the changes in the methods are quite good. However, the authors should check and correct the ecological meaning of the evenness index (l. 228-229).
The paper requires linguistic and grammatical editing (e.g.: l. 32, 33, 86, 89, 166, 171, 172, 176, 263, 247, 268, 375-377 etc, grammar: e.g. l. 263, 268, 310, 313, 314, 315, 330, 331, 333, 426, 436, 441, 446, 447, 444, lack of commas, or dots e.g. l. 89, 329, 330, 331, 332, 461, vocabulary: e.g. l. 166 (lottery=randomly), spelling e.g. 330, 331: grazing, l. 361, 363: roadside etc., correct spelling of units e.g. 100 m, (e.g. l. 171, 172, 358, 359: 0.5325 (?, what?), Latin names of species have to be italic style etc.
There is still the problem with Fig. 1: (A) – is too compressed, (C) is too stretched;
and Fig. 2.: there are still rectangles on B, not squares.
In my opinion in Keywords some words are without sense: abundance, composition, herbaceous, percent cover, richness. Use additionally: invasive species, IAS
The authors have to work with the references, there are still incorrect citations e.g.
l. 673 – the authors of the paper are different then cited: DOI: 10.4236/gep.2017.57001;
l. 675 – different authors,
T., A., F., & A. (2013). Distribution status and the impact of Parthenium weed (Parthenium hysterophorus L.) at Gedeo Zone (southern Ethiopia). African Journal of Agricultural Research, 8(4), 386–397. https://doi.org/10.5897/AJAR12.164
l. 689 – better cite the article than proceedings: https://www.isws.org.in/IJWSn/File/2008_40_Issue-1&2_78-80.pdf
and there is a lot spellings in the citations.
Reviewer 2 Report
The table includes mean values as well as a measure of spread - is it SD? specify.
